# Evaluation of Water and Carbon Estimation Models in the Caatinga Biome Based on Remote Sensing

**Michele L. de Oliveira** [1], **Carlos Antonio Costa dos Santos** [1,2,*], **Francineide Amorim Costa Santos** [2,3], **Gabriel de Oliveira** [4], **Celso Augusto Guimarães Santos** [5], **Ulisses Alencar Bezerra** [6], **John Elton de B. L. Cunha** [6] and **Richarde Marques da Silva** [7]

1   Graduate Program in Engineering and Natural Resources Management, Federal University of Campina Grande, Campina Grande 58109-970, Paraíba, Brazil
2   Academic Unity of Atmospheric Sciences, Federal University of Campina Grande, Campina Grande 58109-970, Paraíba, Brazil
3   Institute of Teacher Training, Federal University of Cariri, Brejo Santo 63260-000, Ceará, Brazil
4   Department of Earth Sciences, University of South Alabama, Mobile, AL 36688, USA
5   Department of Civil and Environmental Engineering, Federal University of Paraíba, João Pessoa 58051-900, Paraíba, Brazil
6   Program in Civil and Environmental Engineering, Federal University of Campina Grande, Campina Grande 58109-970, Paraíba, Brazil
7   Department of Geosciences, Federal University of Paraíba, João Pessoa 58051-900, Paraíba, Brazil
*   Correspondence: carlos.santos@ufcg.edu.br

**Abstract:** The study of energy, water, and carbon exchanges between ecosystems and the atmosphere is important in understanding the role of vegetation in regional microclimates. However, they are still relatively scarce when it comes to Caatinga vegetation. This study aims to identify differences in the dynamics of critical environmental variables such as net radiation (Rn), evapotranspiration (ET), and carbon fluxes (gross primary production, GPP) in contrasting recovered Caatinga (dense Caatinga, DC) and degraded Caatinga (sparse Caatinga, SC) in the state of Paraíba, northeastern Brazil. Estimates were performed using the Surface Energy Balance Algorithm for Land (SEBAL), and comparisons between estimated and measured data were conducted based on the coefficient of determination ($R^2$). The fluxes were measured using the Eddy Covariance (EC) method for comparison with the same variables derived from Moderate Resolution Imaging Spectroradiometer (MODIS) data aboard the Terra satellite. The estimates showed higher Rn values for the DC, indicating that this area should have greater energy availability for physical, biological, and chemical processes. The $R^2$ between daily Rn estimates and observations was 0.93. The ET estimated using the SEBAL showed higher differences in relation to the observed values; however, it presented better spatial discrimination of the surface features. The MOD16A2 algorithm, however, presented ET values closer to the observed data and agreed with the seasonality of the Enhanced Vegetation Index (EVI). The DC generally showed higher ET values than the SC, while the MODIS data (GPP MOD17A2H) presented a temporal behavior closer to the observations. The difference between the two areas was more evident in the rainy season. The $R^2$ values between GPP and GPP MOD17A2H were 0.76 and 0.65 for DC and SC, respectively. In addition, the $R^2$ values for GPP Observed and GPP modeled were lower, i.e., 0.28 and 0.12 for the DC and SC, respectively. The capture of $CO_2$ is more evident for the DC considering the whole year, with the SC showing a notable increase in $CO_2$ absorption only in the rainy season. The GPP estimated from the MOD17A2H showed a predominant underestimation but evidenced the effects of land use and land cover changes over the two areas for all seasons.

**Keywords:** MODIS; net radiation; energy exchange; evapotranspiration; gross primary production; carbon dioxide; semiarid area; eddy covariance

## 1. Introduction

Solar and terrestrial radiation are the primary energy sources for the soil–plant–atmosphere continuum's physical, chemical, and biological processes. The interrelationship between heat, moisture, and radiation exchanges is essential in regulating the global climate [1]. Processes involving land cover changes may affect the climate as surface characteristics influence the atmosphere, thus influencing energy, water, and carbon exchanges [2–5].

Evapotranspiration (ET) and gross primary production (GPP) are critical variables since they represent processes that are directly linked to the coupling between land surface and atmosphere. ET is necessary to understand the dynamics of hydrological processes, which is essential for water management, especially in regions with a semiarid climate [6]. Estimating GPP on a regional scale is still a challenge, and most studies utilize light use efficiency [7]. GPP and ET are the main regulators of water and carbon exchanges between the biosphere and atmosphere.

The terrestrial biosphere influences the surface radiation balance (SRB) and the Earth's climate, mainly affecting atmospheric $CO_2$ and the surface energy balance. The dynamics of the SRB mainly depend on the vegetation type. However, this information is scarce for semiarid regions. Obtaining accurate information about surface energy fluxes in semiarid regions is essential because they have unique characteristics compared to other ecosystems [1]. It should be noted that the semiarid region covers around 11.5% of the Brazilian territory, with Caatinga originally composing ~76% of the Brazilian semi-arid region [1]. The semi-arid areas are characterized by low and irregular rainfall, high temperatures, and solar radiation, which increase evaporation and soil desiccation, leading to water deficits during most of the year [8]. As can be seen, the estimation of water loss becomes relevant in aiding decision-making by stakeholders. Furthermore, environmental degradation potentially affects water and carbon fluxes [5,9]. Monitoring and detecting environmental degradation-prone areas are just the first steps in providing valuable information for decision-makers to mitigate the effects and prevent climate disasters by the establishment of policies regulating land exploration and occupation [3].

According to the authors [10], among all Brazilian biomes, the greatest interest in studying carbon sequestration is directed to the Amazon Forest. Despite being one of the most threatened biomes due to hundreds of years of inefficient and unsustainable use of soils and natural resources, the Caatinga is the most neglected Brazilian biome in the most different sense. This problematic situation has come to be considered by various governmental and non-governmental sectors. In addition to the great need to preserve their natural systems, a serious insufficiency of scientific knowledge is still a challenge [11,12]. Despite different studies carried out in the Caatinga region, there is much to be understood regarding the turbulent surface fluxes. It remains a challenge to understand how meteorological and environmental variables influence energy and mass fluxes at the biosphere-atmosphere interface, particularly in cases of suppression of native vegetation and the associated changes in the microclimate [1]. Despite the fact that Caatinga is an endemic Brazilian biome with expressive vegetation heterogeneity, few studies evaluate and validate data sets from the Moderate Resolution Imaging Spectroradiometer (MODIS). The MODIS products use simplified assumptions in algorithms to describe the processes of water and carbon [12–15].

MODIS products are available in a cumulative composition of 8-day values with a pixel size of 500 m [16]. Several studies have evaluated the performance of MODIS products in recent years for different land covers [17–20]; however, the validation of these products for the Caatinga biome is still lacking. In addition to the validation of MODIS products, this study evaluates two other models: (i) the Surface Energy Balance Algorithm for Land (SEBAL) for ET estimates, and (ii) the [21] modeling approach for GPP estimates.

SEBAL estimates ET as the residual of the surface energy balance. The main component of the algorithm is the internal process of estimating the near-surface (dT) temperature gradient, which is based on the selection of endmembers that represent the extremes of the

wet (cold) and dry (hot) ET spectrums [22,23]. The modeled GPP was based on the concept of radiation use efficiency [21].

According to [24], the Brazilian semiarid region is one of Brazil's most vulnerable regions, from a social perspective to climate change. Within this scope, it is essential to explore how the replacement of natural vegetation in the Caatinga, as well as its degradation, can cause changes in the energy, water, and carbon fluxes in this region. Different studies have been conducted to estimate energy, water, and carbon fluxes between semiarid ecosystems and the atmosphere, generally using the eddy covariance technique [1,5,24] and remote sensing data [6,7,12,25–27]

The present study has the main goal of evaluating different models for estimating water and carbon fluxes in areas of Caatinga with different levels of degradation (dense and sparse) in the state of Paraíba, Brazil, using orbital remote sensing data.

## 2. Material and Methods

### 2.1. Study Area

The study area is located in the state of Paraíba, northeastern Brazil. Figure 1 shows the land use and land cover (LULC) for the state of Paraíba in the year 2020, and the location of the two micrometeorological towers used in this study. In the study area, the dominant soil type is Litholic Neosol, characterized as shallow, with a maximum depth of 10 cm, excessively drained, stony, and gently wavy [28]. Based on the Köppen classification, the region's climate is semiarid (type BSh) with low latitude and altitude [29]. The rainy season starts in February/March and lasts until July/August. The average annual rainfall is ~783 mm·year$^{-1}$, from 1935 to 2014 [5].

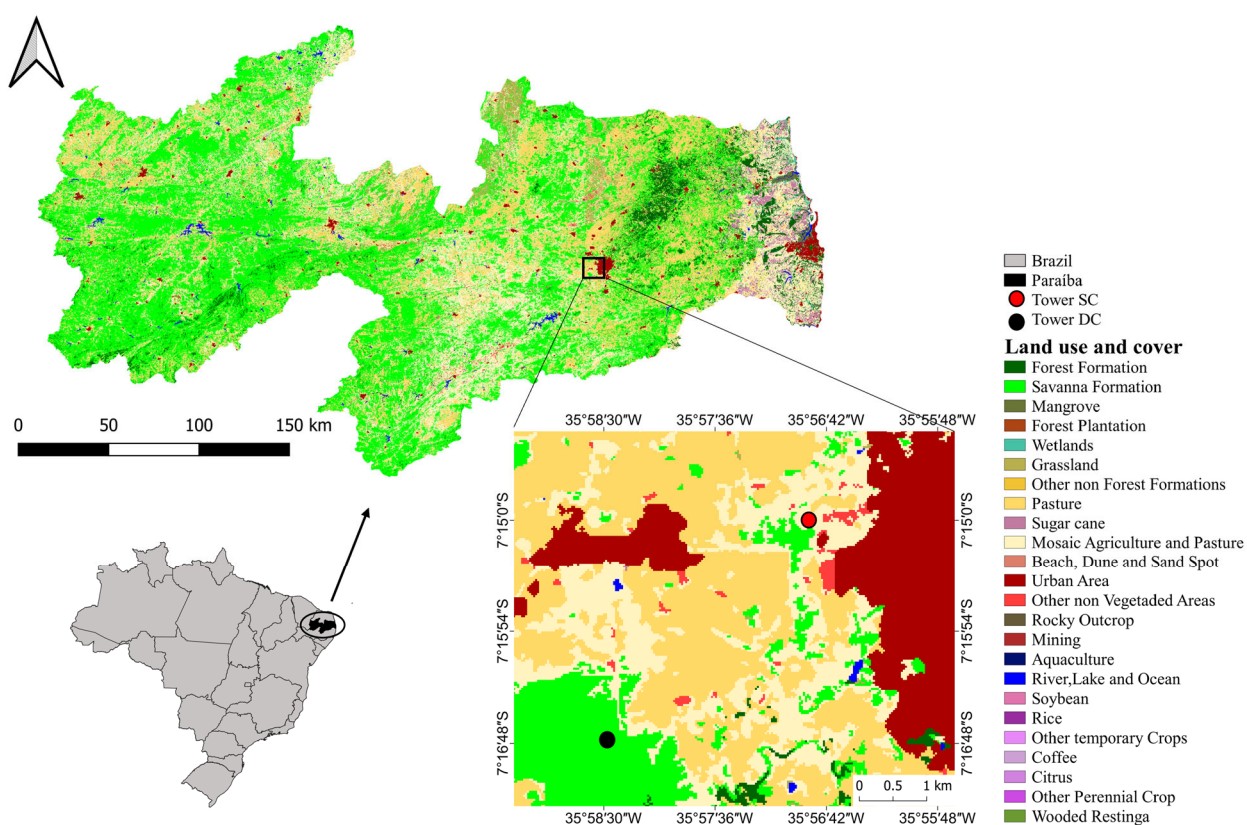

**Figure 1.** Land use and cover map of the study area and location of micrometeorological towers.

### 2.2. Ground Measurements

The measured data were obtained via two micrometeorological towers installed approximately 4.5 km apart at the Instituto Nacional do Semiárido (INSA), located in Campina

Grande. The database used in this study is from 2013 and was used to validate the energy, water, and carbon fluxes estimate based on remote sensing estimates. The eddy covariance towers are located over different Caatinga areas with contrasting characteristics, i.e., one tower is located in a dense area of Caatinga (DC), and the second is installed in a sparse area of Caatinga (SC).

The DC study area has 675 ha, of which approximately 300 ha are well preserved in various stages of development. It is an area covered by sub-deciduous and deciduous forests, which lose their leaves in the driest months and are composed mainly of arboreal and shrub species. The trees have an average height of ~5.1 m. The SC area is an anthropized area, located approximately 1 km from the urban area of the municipality of Campina Grande. The SC area has sparse vegetation, with a predominance of shrubby Caatinga, composed of shorter trees, up to 2 m in height, and the presence of bare soil and rocky outcrops [1]. The measurements of environmental variables were carried out using sensors installed in the two towers at a height of 2 m above the vegetation's canopy. The footprint of the towers is approximately 1 km, enough to capture vegetation heterogeneity with a predominance of *Mimosa hostilis* trees, as described in [1].

The data were stored in a Datalogger CR3000 (Campbell Scientific), with the average values stored at each half-hour interval. The three-dimensional fluctuation of wind speed was measured using a sonic anemometer (CSAT3A, Campbell Scientific). The fluctuations in water vapor and carbon dioxide concentration were measured using a gas analyzer (EC150, Campbell Scientific). The measurements were obtained at a frequency of 10 Hz. The radiation balance components were quantified with a radiometer balance (CNR 4, Kipp & Zonen); air temperature (HC2S3-L, Campbell Scientific), with measurements recorded every 5 s, with their average and total values stored every 30 min. Following the collection of the high-frequency data, the averages for half-hour intervals were calculated. Energy, water, and carbon fluxes were quantified using the eddy covariance technique, as described in [1,30]. The detailed installation of the instruments is described by [9]. In addition to periodic instrument maintenance and calibration, the data underwent a rigorous post-processing process. Detailed information on data processing, quality control, and post-processing can be found in [1]. Details on obtaining water (ET) and carbon (GPP) fluxes from flux towers are described in [5].

### 2.3. Remote Sensing Data

The remote sensing data were obtained through MODIS products for the year 2013. The following MODIS products were used: MOD11A2 (surface temperature), MOD09A1 (surface reflectance), MOD07_L2 (air temperature and dew point), MOD05 (precipitable water), MOD03 (geolocation), MCD12Q1 (ground cover), MOD13Q1 (vegetation index), MOD16A2 (ET), and MOD17A2H (GPP) (https://modis.gsfc.nasa.gov/data (accessed on 23 October 2022)).

The spatial subset for the remote sensing estimates was 1 km around each tower, which corresponds to a $3 \times 3$ window, totaling 9 pixels. We performed analyses in relation to the footprint between the towers and the remote sensing data and observed that the subsetting at 1 km showed more similar results. Although the footprint does not cover a completely homogeneous region, they are representative samples as we are monitoring areas that correspond to the same elements, with a greater proportion of this portion (greater than 90%) being the focus of our study (DC and SC).

### 2.4. Data Processing and Analysis

2.4.1. Atmospheric Parameters (Precipitable Water and Air Temperature) and Daily $R_n$ Estimation

The estimation of the instantaneous Rn was conducted using only MODIS data, aiming to develop an operational methodology. We understand that this approach can facilitate future analysis in similar ecosystems, given the significant lack of Rn estimates globally [31]. Rn is not usually measured at weather stations, meaning that its estimation is

often restricted to experimental campaigns, which are more costly. The instantaneous Rn was used to estimate the evaporative fraction and the daily Rn to calculate the actual daily evapotranspiration (ET).

The first step is to estimate the instantaneous $R_n$ as follows:

$$Rn = Rs(1 - \alpha) + R_{L\downarrow} - R_{L\uparrow} - (1 - \varepsilon_0)R_{L\downarrow} \tag{1}$$

where $R_s$ (W·m$^{-2}$) is the downwelling shortwave radiation; $\alpha$ (dimensionless) is the surface albedo; $R_{L\downarrow}$ (W·m$^{-2}$) is the downwelling longwave radiation; $R_{L\uparrow}$ (W·m$^{-2}$) is the upwelling longwave radiation; $(1 - \varepsilon_0)\,R_{L\downarrow}$ (W·m$^{-2}$) represents the fraction of incident longwave radiation reflected by the surface; and $\varepsilon_0$ is the surface emissivity (dimensionless). The term $\alpha\,R_s$ (W·m$^{-2}$) represents the shortwave radiation flux reflected by the surface (upwelling shortwave radiation). More detailed information about this methodology can be found in [32].

To investigate the impact of considering the water content in the atmosphere on the $R_n$ estimates, our methodology aimed to generate the $R_n$ spatial distribution in two ways: (a) using Equation (2), in order to estimate $R_n$ 1 (where the transmissivity only depends on altitude) [33], and (b) using Equation (3), to estimate $R_n$ 2 (where the transmissivity considers the water content in the atmosphere) [22]:

$$\tau_{sw} = 0.75 + 2 \cdot 10^{-5}z \tag{2}$$

where z represents each pixel's altitude in the image, obtained from the digital elevation model generated by the Shuttle Radar Topography Mission (SRTM), available at http://srtm.csi.cgiar.org (accessed on 23 October 2022).

$$\tau_{sw} = 0.35 + 0.627 \exp\left[\frac{-0.00146P}{K_t \cos \theta_z} - 0.075\left(\frac{WP}{\cos \theta_z}\right)^{0.4}\right] \tag{3}$$

where $\theta_z$ is the solar zenith angle, obtained from the product MOD09A1, P is the average atmospheric pressure in kPa, $K_t$ is the atmospheric turbidity coefficient, set to 1 for a clear sky and set to 0.5 for extreme turbidity [34], and WP is the precipitable water, obtained from the MOD05 product.

For validation purposes and aiming at greater operability when using only satellite data, a comparison was made between the WP values obtained by the MOD05 product and those estimated from surface data using the following equation [35]:

$$WP = 0.14e_a P_{air} + 2.1 \tag{4}$$

where $e_a$ is the near-surface vapor pressure (KPa), $P_{air}$ is the atmospheric pressure, both obtained from surface data obtained according to [36].

The average net radiation that occurred in the 24-h period ($R_{n24h}$, in W·m$^{-2}$, is obtained using Equation (5), developed by Slob [37]:

$$Rn_{24h} = (1 - \alpha)R_{S\downarrow24h} - a\tau_{sw24h} \tag{5}$$

where $\alpha$ is the surface albedo, $R_{S\downarrow24h}$ is the daily average incident solar radiation (W·m$^{-2}$), and a is the regression coefficient between the daily longwave net radiation and the daily atmospheric transmissivity. The surface albedo is assumed to be equal to the albedo during the satellite overpass time, estimated as proposed by [38]. The daily solar radiation and transmissivity were obtained from ground measurements [39]. The coefficient a was set to 110, according to [39], and then, for comparison purposes, the value of 98.208 was used, calibrated by [40] for the northeastern region of Brazil.

The daily average atmospheric transmissivity was obtained as:

$$t_{sw24h} = \frac{R_{S\downarrow 24h}}{R_{S\downarrow TOA24h}}$$

(6)

where $R_{S\downarrow TOA}$ is the daily solar radiation incident at the top of the atmosphere (W·m$^{-2}$), calculated according to procedures described in [41].

2.4.2. ET (SEBAL versus MOD16A2) Estimation

Energy balance models based on remote sensing data have been widely used to estimate agricultural water use and evapotranspiration and map the spatial distribution of energy fluxes across landscapes [42]. In this sense, the SEBAL algorithm developed for estimating ET using satellite images has been validated in several experimental campaigns throughout the world [43–45].

To estimate ET on a daily scale, it is assumed that the instantaneous evaporative fraction (estimated from the energy balance components) is constant for 24 h. Refs. [39,46] stated that the evaporative fraction is a relatively constant indicator of energy partitioning at the surface during daylight hours. It has a minimal diurnal cycle, improving the usefulness of instantaneous satellite imagery for studying the interactions between the land surface and the atmosphere. The energy balance components for the surface were then estimated from:

$$Rn - G - LE - H = 0$$

(7)

where $R_n$ is the net radiation (Equation (1)), G is the soil heat flux, LE is the latent heat flux, and H is the sensible heat flux, all in units of W·m$^{-2}$.

Soil heat flux (G) was estimated according to the equation described in [39]:

$$\frac{G}{R_n} = \frac{T_s}{\alpha}\left(0.0038\alpha + 0.0074\alpha^2\right)\left(1 - 0.98NDVI^4\right)$$

(8)

where $T_s$ is the surface temperature, $\alpha$ is the surface albedo, $R_n$ is the net radiation, and NDVI is the Normalized Difference Vegetation Index, as described next. For water bodies, the heat flux was taken as 30% of $R_n$ [47].

The NDVI is the ratio between the differences in the reflectivity of the near-infrared ($\rho_{IV}$) and red ($\rho_V$) bands and their sum; for the MODIS/Terra sensor, $\rho_{IV}$ corresponds to channel 2 and $\rho_V$ to channel 1:

$$NDVI = \frac{\rho_{IV} - \rho_V}{\rho_{IV} + \rho_V}$$

(9)

Sensible heat flux (H) was estimated according to [23]:

$$H = \rho c_p \frac{dT}{r_{ah}}$$

(10)

where $\rho$ corresponds to the air-specific mass, $c_p$ is the air specific heat at constant pressure (1004 J·kg$^{-1}$·K$^{-1}$), dT is the temperature difference between two levels $Z_1$ and $Z_2$, and $r_{ah}$ is the aerodynamic resistance to the transport of heat (s·m$^{-1}$).

Based on $R_n$, G, H, and LE (W·m$^{-2}$), the evaporative fraction ($\Lambda$ is estimated according to:

$$\Lambda = \frac{LE}{R_n - G}$$

(11)

The actual daily evapotranspiration (mm·day$^{-1}$) was estimated using:

$$ET_{24h} = \frac{86400\Lambda R_{n24h}}{\lambda\rho_w}$$

(12)

where $\lambda$ is the latent heat of water vaporization ($J \cdot kg^{-1}$), estimated according to [48] through Equation (13); $86{,}400/\lambda$ factor is used to convert ET from $W \cdot m^{-2}$ to $mm \cdot day^{-1}$; $\rho_w$ is the water density ($kg \cdot m^{-3}$).

$$\lambda = [2.501 - 0.00236(T_a - 273.16)] * 10^6 \tag{13}$$

where $T_a$ is the air temperature obtained from the MODIS product, in K. $\lambda$ values vary slightly over normal temperature ranges. A single value can be adopted in many cases, e.g., for T = 20 °C, $\lambda$ can be set to 2.45 $MJ \cdot kg^{-1}$ [33].

The evapotranspiration values obtained from the MOD16A2 products were extracted for comparison purposes. The ET was estimated according to the methodology described by [49], which uses satellite-derived energy fluxes and the leaf area index based on the Penman-Monteith method and later improved by [50]. MOD16A2 provides, among other estimates, 8-day periods of accumulated ET with a spatial resolution of 500 m.

2.4.3. Gross Primary Production (GPP) (Modeled versus MOD17A2H) Estimation

GPP is the total carbon assimilated by vegetation, of which a part is lost to the atmosphere because of autotrophic respiration. Thus, GPP is the primary regulator of carbon exchange from the atmosphere to the Earth. Consequently, a better understanding of the spatiotemporal dynamics of GPP provides a valuable measure of ecosystem health and the impacts of regional LULC disturbances and climate change, thus improving carbon cycle estimates [12].

In this study, the GPP product from MOD17A2H (called GPP MOD17A2H) was used. This product was based on [21], which relates gross photosynthesis to the amount of photosynthetically active radiation (PAR) absorbed by photosynthetic biomass and a conversion efficiency term [51]:

$$GPP = \varepsilon \times fPAR \times PAR \tag{14}$$

where $\varepsilon$ is the radiation use efficiency (RUE) of vegetation ($kg \cdot C \cdot MJ^{-1}$), PAR is the photosynthetically active radiation ($MJ \cdot d^{-1}$), and fPAR is the fraction of incident PAR absorbed by the canopy. Equation (14) was also used to estimate the GPP on a smaller scale (GPP Modeled), as described in the following:

The fPAR estimate was obtained according to [52] as follows:

$$fPAR = (-0.161 + 1.257 NDVI) \tag{15}$$

The acquisition of MODIS products was performed using the Google Earth Engine (GEE) platform, available at https://code.earthengine.google.com (accessed on 23 October 2022). The $\varepsilon$ parameter was estimated using:

$$\varepsilon = \varepsilon_{max}.T_1.T_2.\Lambda \tag{16}$$

where $\varepsilon_{max}$ is the maximum efficiency of light use by vegetation, whose value was set to 2.15 $g \cdot C \cdot MJ^{-1}$ [21,53]). $T_1$ and $T_2$ are estimated as follows:

$$T_1 = 0.8 + 0.02 T_{opt} - 0.005 T_{opt}^2 \tag{17}$$

$$T_2 = \frac{\frac{1}{1+\exp\left(0.2 T_{opt} - 10 - T_{day}\right)} * 1}{1 + \exp\left[0.3\left(-T_{opt} - 10 + T_{day}\right)\right]} \tag{18}$$

where $T_{opt}$ and $T_{day}$ are, respectively, the average air temperature during the maximum leaf area index, or maximum NDVI, and the daily average temperature. The $T_1$ factor is essentially responsible for reducing the cooler regions' effect on plant growth. In contrast,

the $T_2$ factor reduces the light use efficiency ($\varepsilon$) if the ambient temperature deviates from the optimal temperature, which is relevant for arid and semiarid regions.

To estimate PAR, a model modified by [52] and applied in the Northeastern region of Brazil by [53] was used. It combines the model proposed by [21], which is based on PAR, with the light use efficiency model of [54] and the energy balance model of [23]. Thus, PAR is estimated as:

$$PAR = 0.48.R_{s24h} \tag{19}$$

with $R_{s24h}$ in W·m$^{-2}$.

*2.5. Data Analysis*

For the validation and analysis of the observed and estimated results in 2013 in the DC and SC areas, the mean absolute error (MAE), mean percent error (MPE), coefficient of determination ($R^2$), correlation coefficient (r), and root mean square error (RMSE) were used and obtained as:

$$MAE = \frac{1}{N}\sum_{i=1}^{N}|X'_i - X_i| \tag{20}$$

$$MPE = \frac{100}{N}\sum_{i=1}^{N}\left|\frac{X'_i - X_i}{X_i}\right| \tag{21}$$

$$RMSE = \sqrt{\frac{1}{N}\sum_{i=1}^{N}(X'_i - X_i)^2} \tag{22}$$

where X corresponds to the observed value, X$'$ to the estimated value, and N the amount of data analyzed.

For the spatial analyses, the annual cumulative, standard deviation (SD), maximum (max), and median of ET and GPP in the state of Paraíba in 2013 were calculated for the different models.

## 3. Results and Discussion

*3.1. Validation of Atmospheric Parameters (Precipitable Water and Air Temperature)*

Table 1 shows the average values of precipitable water estimated MOD05 and those obtained in situ in the DC area. We note that the observed values were obtained using Equation 4 together with field vapor pressure and atmospheric pressure data. The measured data in SC showed gaps. For this reason, the model was not evaluated using SC data.

For the DC area, the mean observed value was 29.5 mm and estimated at 29.8 mm, with a minimum percentage error of 1.5% and a maximum of 35.5%, resulting in an average percentage error of 15.2%. Here, we emphasize the importance of obtaining an estimation based only on orbital data with relatively high precision. Moreover, when it comes to operability, an important feat with satellite-derived products is accurate precision in air temperature estimates. This is important, especially due to the fact that this variable is practically restricted to data from meteorological stations. Thus, it is valuable to validate estimates based on remote sensing data. We used data from the MOD07_L2 product as a substitute for the air temperature. This product provides the inference of the atmospheric profile at 20 pressure levels, from 5 to 1000 hPa. For further details, see [55].

The air temperature values measured in the towers and estimated from MOD07_L2 for the two study areas are also shown in Table 1. For DC, the MPE was 9.4%, and for SC, it was 8.0%, which is relatively small, especially when taking into consideration the advantage of not using point measurements of temperature in place of areal measurements, which represent all the heterogeneity of an area. In this regard, an overestimation was observed in comparison to the measured data. The estimated values showed the same pattern as those measured, following the respective increases or decreases in values. There was a closeness between the air temperatures in the two areas for most of the study days. The average air temperature obtained from MODIS data was the same for the two areas (25.9 °C), while the

field measurements resulted in an average temperature of 28.1 °C for DC and 27.2 °C for SC, respectively. These results are similar to those found by [56] comparing remote sensing air temperature estimates with eddy flux towers in the Brazilian Amazon. The authors found errors between 3.7 and 6.6%.

**Table 1.** Statistical analyzes for Precipitable Water (WP) in mm, Air Temperature (Tar), in °C, Instantaneous Net Radiation ($R_n$ 1 and $R_n$ 2), in $W \cdot m^{-2}$, and Daily Net Radiation ($R_n$24 A and $R_n$24 B), in $W \cdot m^{-2}$, for the year 2013 in the DC and SC areas.

| | **DC** | | | | | |
| | **Average $\pm$ SD** | | | | | |
| | **Observed** | **Estimated** | **$R^2$** | **MAE** | **MPE** | **RMSE** |
| WP | $29.5 \pm 5.1$ | $29.8 \pm 5.5$ | 0.05 | 4.79 | 0.15 | 6.47 |
| Tar | $28.1 \pm 1.9$ | $25.9 \pm 2.6$ | 0.54 | 2.62 | 0.09 | 2.94 |
| $R_n$ 1 | $614 \pm 57.8$ | $663.1 \pm 63.2$ | 0.02 | 67.47 | 0.11 | 82.63 |
| $R_n$ 2 | | $647.7 \pm 67.7$ | 0.03 | 61.37 | 0.10 | 76.75 |
| $R_n$24 A | $186.6 \pm 16.1$ | $164.6 \pm 21.4$ | 0.93 | 24.11 | 0.13 | 24.89 |
| $R_n$24 B | | $157.5 \pm 21.1$ | 0.93 | 31.18 | 0.17 | 31.72 |
| | **SC** | | | | | |
| | **Average $\pm$ SD** | | | | | |
| | **Observed** | **Estimated** | **$R^2$** | **MAE** | **MPE** | **RMSE** |
| Tar | $27.2 \pm 1.9$ | $25.9 \pm 2.7$ | 0.61 | 2.13 | 0.08 | 2.34 |
| $R_n$ 1 | $655.1 \pm 45.3$ | $589.4 \pm 63.8$ | 0.14 | 74.17 | 0.11 | 86.51 |
| $R_n$ 2 | | $575.1 \pm 65.2$ | 0.12 | 82.78 | 0.13 | 97.66 |
| $R_n$24 A | $188.1 \pm 19.2$ | $149.2 \pm 20.4$ | 0.73 | 42.76 | 0.23 | 44.01 |
| $R_n$24 B | | $142.1 \pm 20$ | 0.76 | 49.82 | 0.27 | 50.78 |

### 3.2. Instantaneous Net Radiation Validation

Table 1 shows the validation of the instantaneous Rn estimated using the two methodologies (Rn 1 and Rn 2) in relation to the observed values. Although the applied methodologies overestimated the field measurements on most days in DC, it is interesting to emphasize the role of the MOD05 product in the results, which caused attenuation in the Rn values. The methodology's performance corroborates other studies, such as those by the authors of [57], who found RMSE values ranging from 61.10 to 82.83 $W \cdot m^{-2}$ when estimating net radiation exclusively from MODIS at eight different locations over the continental United States.

The data showed higher values for the SC than for the DC when considering the observed Rn. Observing other values found in the literature leads us to believe that the measurements performed in SC must have had some interference that caused an increase in the observed Rn, making it greater than that in DC. We note that the expected behavior was that higher values of Rn would occur in the dense area in comparison to the sparse area. The land covered by dense vegetation is expected to retain a greater amount of energy. It reflects less incoming radiation (lower albedo), indicating greater energy available to be converted into sensible and latent heat than the sparse area [1]. This can be observed in the results obtained via orbital data, with higher Rn values in DC than in SC. The authors of [1] showed that, in the SC area, the incidence of solar radiation was higher. These results indicate that the high solar radiation in this site may be a crucial contributing factor to the higher values of Rn.

These results corroborate those of [24] in the municipality of Petrolina, state of Pernambuco based on simulations of land cover change with the Integrated Biosphere Simulator

(IBIS) model. The authors found a decrease in Rn of 35% as a result of the conversion of natural vegetation from Caatinga to Caatinga sparse in relation to the annual average of Rn for the preserved Caatinga. The authors of [58], using Landsat images in a region occupied by dense savanna in the São Francisco River Basin, found Rn values of 680.0 to 732.0 W·m$^{-2}$. The areas with open steppe savanna, on the other hand, presented Rn values predominantly between 622.0 and 680.0 W·m$^{-2}$. In the same study and for the same date, the use of the MODIS/Aqua sensor revealed values of around 613.0 to 669.0 W·m$^{-2}$ in the dense savanna region; for an area of exposed soil or very sparse native vegetation, estimates were in the range of approximately 510.0 to 572.0 W·m$^{-2}$.

For the present study, the average Rn observed was 614 W·m$^{-2}$ and 655.1 W·m$^{-2}$ in the DC and SC areas, respectively. With estimated average Rn 1 and Rn 2 of 663.1 W·m$^{-2}$ and 647.7 W·m$^{-2}$ for the DC area and 589.4 W·m$^{-2}$ and 575.1 W·m$^{-2}$ for the SC area. The authors of [25], in a study conducted in the Seridó ecological station, obtained values between 500 and 700 W·m$^{-2}$ for the Rn estimated exclusively via MODIS data. This corroborates with the results presented in Table 1 for DC.

### 3.3. Daily $R_n$ Validation

To estimate and validate the daily net radiation (Rn24), the measured values of incoming short-wave radiation were used. Using measured data in the estimate does not reduce the method's operability, as these values are available from meteorological stations. It is not necessary to set up an experiment to use them. The results are shown in Table 1. The two estimates are presented: Rn24 A, which used a value of 98.208, calibrated by the authors of [40] for northeastern Brazil, and Rn24 B, which uses the value of a (in Equation (5)) equal to 110, according to [39]. The authors of [40], in a study conducted in the Quixeré region, state of Ceará, using Landsat 5/TM images, observed MPE values of the order of 7.69% and 9.11% for Rn24 A and Rn24 B, respectively. Here, the MPE for DC was 13.2% for Rn24 A and 17.0% for Rn24 B. [59], in a study over a cerrado area in São Paulo State, found an MPE of 10.6% when using Rn24 B. The authors of [56] observed errors between 12.5% and 16.4% and 11.3% and 15.9% for instantaneous and daily Rn, respectively, by comparing MODIS and eddy flux towers in the Brazilian Amazonia.

Again, greater energy availability in DC is observed from the results of the estimates, in agreement with the instantaneous Rn values. However, instantaneous radiation balance estimates are necessary for some studies. Nevertheless, they do not have as much practical applicability as the daily Rn since they estimate evapotranspiration from remote sensing data [57]. Table 1 shows the coefficient of determination ($R^2$) between estimates and observations, and despite Rn24 A having a lower MPE, Rn24 B showed a better correlation with the observed data. This result was crucial to determine the choice of the latter to estimate daily evapotranspiration in the following sections.

However, the estimates underestimated the field measurements, a result contrary to the one obtained for the Caatinga area by [25]. They used the sinusoidal model to extrapolate the instantaneous values of Rn on a daily scale. Here, it is important to note that in [25], the surface temperature was used as a substitute for the air temperature in the computation of the downwelling longwave radiation. This component of the radiation balance may be responsible for an increase in the Rn value.

### 3.4. ET Validation

As a result, Table 2 shows the ET estimated according to Equation (12), here called ET SEBAL, and the ET according to the MOD16A2 product was underestimated in relation to the observed data (EC technique), except for ET SEBAL in DC, which overestimated the observed values and MOD16A2 in the dry period, as can be seen in Figure 2. The absolute differences, however, were greater for ET SEBAL (Table 2). This result may be due to an apparent underestimation in the estimated H values (not shown), which indicates that the rain occurred four days before the image was acquired for almost all the dates studied.

**Table 2.** Statistical analyzes for ET (MOD16A2 and SEBAL), in mm·d$^{-1}$, and GPP (MOD17A2H and Modeled), in g·C·m$^{-2}$·d$^{-1}$, for the year 2013 in the DC and SC areas.

| | | | | | | |
|---|---|---|---|---|---|---|
| **DC** | | | | | | |
| | **Average ± SD** | | | | | |
| | **Observed** | **Estimated** | **R$^2$** | **MAE** | **MPE** | **RMSE** |
| ET MOD16A2 | 2.16 ± 1.49 | 1.91 ± 1.3 | 0.67 | 0.73 | 1.19 | 0.87 |
| ET SEBAL | | 2.61 ± 0.42 | 0.30 | 2.01 | 3.67 | 2.19 |
| GPP MOD17A2H | 8.58 ± 5.0 | 3.68 ± 1.54 | 0.76 | 4.04 | 0.50 | 4.9 |
| GPP Modeled | | 6.69 ± 2.02 | 0.28 | 3.19 | 0.28 | 4.84 |
| **SC** | | | | | | |
| | **Average ± SD** | | | | | |
| | **Observed** | **Estimated** | **R$^2$** | **MAE** | **MPE** | **RMSE** |
| ET MOD16A2 | 2.39 ± 1.12 | 1.22 ± 0.67 | 0.66 | 0.60 | 0.52 | 0.74 |
| ET SEBAL | | 1.81 ± 0.51 | 0.48 | 1.26 | 1.17 | 1.50 |
| GPP MOD17A2H | 3.42 ± 1.64 | 2.56 ± 1.25 | 0.65 | 2.32 | 0.47 | 2.60 |
| GPP Modeled | | 2.63 ± 1.96 | 0.12 | 2.01 | 0.50 | 2.26 |

Figure 2 shows ET SEBAL and ET MOD16A2 in relation to field measurements. Except for the ET SEBAL estimated in the DC area, the seasonality of ET is well-pronounced, especially for the rainy season (March to July). Emphasizing the ET MOD16A2, we observed values closer to the measured ET than the ETSEBAL. Although the MODIS data underestimated the observed data, these estimates were able to follow the seasonality of the study areas, with higher precision for the DC area.

The low performance of the MODIS estimates in the SC area is related to the fact that the region has large portions of exposed soil, which makes the performance of the MODIS LAI algorithm difficult. This algorithm presents higher precision on dense vegetation and in areas with less soil influence, thus interfering with the MOD16A2 estimates. As described in [25], the amplitude of hot or cold pixels in an image, which is very subtle over drylands in dry conditions, is known to affect SEBAL. The hottest and coldest pixels are given the maximum and minimum sensible heat fluxes, respectively. Due to the inability to distinguish between the region's land cover, especially during the dry season, and the fact that bare soil and senesced vegetation have similar LAI values, the 500 m spatial resolution of MODIS images also makes it difficult to distinguish between hot and cold pixels.

Regarding the evidence of seasonality, the authors of [25] found a good result for applying the SEBAL algorithm. However, that did not occur for the validation using field measurements. In that study, the estimate using ET MOD16A2 showed good performance in obtaining the spatial and temporal variability of water exchange in the region, which is in agreement with the results obtained here. The ET MOD16A2 also showed a better relationship with the EVI (Figure 2), indicating the differences between the two study areas. The relationship of ET with the increase in the vegetation index (EVI) is also pointed out by [7] in a study conducted in the Ziya-Daqing basin, China, using a model for estimating ET (the VIP model, Vegetation Interface Processes). They used the ET of the MODIS product (MOD16A2) compared to the ET measured by EC for different types of ground cover. Several studies have pointed for an increase in the vegetation index in semiarid regions causes an increase in ET [7,25,60].

The DC presented higher EVI values (reaching 0.6) than the SC (a minimum of 0.11), although both are dependent on the rainfall regime, which is more intense during the rainy season (Figure 2). This result corroborates a study carried out [61] in dense and open Caatinga areas, where they found values around 0.9 for dense Caatinga in the rainy season. In contrast, the values for open Caatinga were lower than 0.3 in the dry period.

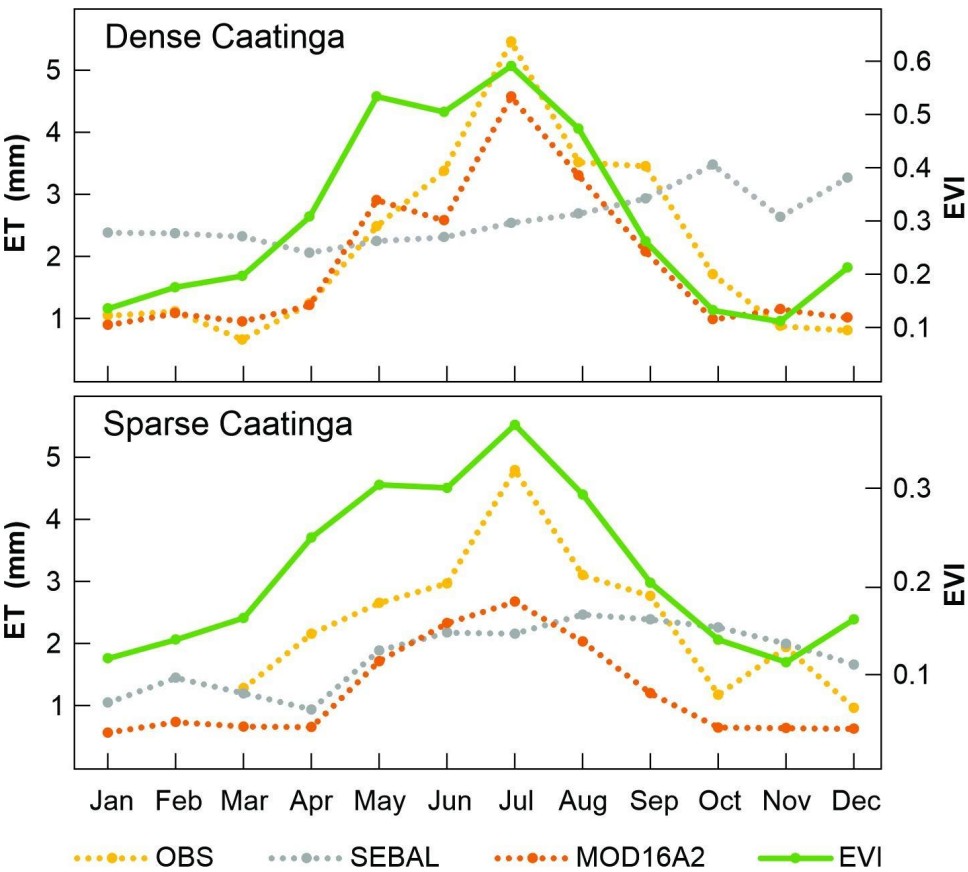

**Figure 2.** Monthly temporal distribution of ET in the areas of DC and SC in 2013.

These results agree with those observed in [58], who found that the actual daily evapotranspiration (ET) values for the DC region ranged from 5.0 to 6.7 mm·day$^{-1}$ decrease in the ET values when the native vegetation becomes sparser. In areas with open or semi-open savannah, ET values were between 4.0- and 5.0-mm·day$^{-1}$, and observed values ranged from 3.0 to 4.0 mm·day$^{-1}$. In Table 2, the mean values of the ET observed in DC were 2.2 mm·day$^{-1}$ (ranging from 0.11 to 4.8 mm·day$^{-1}$), from 2.6 mm·day$^{-1}$ for the ET SEBAL (ranging from 2.3 to 3.5 mm·day$^{-1}$) and 1.9 mm·day$^{-1}$ for MOD16A2 ET (ranging from 0.4 to 5.8 mm·day$^{-1}$). For SC, the mean value was 2.4 mm·day$^{-1}$ for the measured ET (ranging from 0.3 to 4.1 mm·day$^{-1}$), 1.8 mm·day$^{-1}$ for the ET SEBAL (ranging from 1.6 to 3.6 mm·day$^{-1}$), and 1.2 mm·day$^{-1}$ for the ET of MOD16A2 (ranging from 0.4 to 3.3 mm·day$^{-1}$).

It is observed that, considering the average values, the results of MOD16A2 showed very good agreement with the field measurements. This evidence shows that our methodology is a reliable alternative and simple to perform for estimating ET in the Caatinga biome, both for the dry and rainy seasons, corroborating previous results [25].

The authors of [6], in a study carried out in the Caatinga area in the municipality of Petrolina using two Landsat 8 images, found maximum Eta values of 8.0 mm·day$^{-1}$ using the SEBAL method (SEBAL manual). In the same study, the authors used, for comparison purposes, the Geographic Resources Analysis Support System (GRASS) Python surface energy balance algorithm for land (GP-SEBAL), a new Python-based methodology for applying fully automated versions of SEBAL to Landsat images, obtaining a maximum value of 7.6 mm·day$^{-1}$. The results differ from this study; however, in addition to the specificities of each study area and the differences inherent to the sensors used, it is necessary to highlight that our study calculated the latent heat of vaporization based on the air temperature product from MODIS (MOD07_L2). In contrast, the study above used a constant value for this same variable. Moreover, using an automatic method but applied

to MODIS images, the automated surface energy balance algorithm for land (ASEBAL), proposed by [26], estimated the ET for 282 images in the Caatinga area in the Ipanema River basin, with values ranging from 1.6 mm·day$^{-1}$ to 6.2 mm·day$^{-1}$ and a mean ET of 3.9 mm·day$^{-1}$, which is closer to our results.

The best performance for the ET of the MOD16A2 is related to the results obtained for the two control points within the study areas. This is a coherent result, as the eastern portion is closer to the Zona da Mata, which has a more humid climate with a predominance of Atlantic Forest vegetation. The analysis of ET values is critically related to understanding the characteristics of the vegetation at the satellite overpass time [6]. This understanding may be essential in the management of water resources and in encouraging the preservation of native vegetation. Furthermore, ET estimation can assess ecosystem losses at climatic extremes, which is vital for decision-making policy reports [25,62].

### 3.5. Gross Primary Production (GPP) Validation

The GPP from the EC (observed), the modeled GPP, and the GPP extracted from MOD17A2H are shown in Figure 3. It is observed that there is a greater difference in GPP values between the two areas during the rainy season as a response to greater rainfall availability (Figure 3). In other periods, which are drier, DC captures $CO_2$ closer to SC. It was expected that this difference would be less noticeable in the dry season due to the loss of leaves (senescence) [5]. It shows that rain is the limiting factor in $CO_2$ capture since the smallest difference between GPP in the two areas occurred in the period with less water availability (August to February). This result corroborates previous results [13]. The authors observed that the difference between the GPP estimated and the measured data is much greater in the wet season than in the dry season. Even though MOD17A2H GPP greatly underestimates the measured data, it manages to differentiate the seasonal behavior of the two study areas. It is worth mentioning that this estimate was made on an 8-day basis, and the value presented here corresponds to the average for the period.

In a study carried out in northern Australia, the authors of [63] found an R2 between GPP measured and GPP_MOD17 for savanna vegetation equal to 0.32. As for the Caatinga area, the authors of [13] obtained a reasonable value of 0.43 for $R^2$. In this study, the $R^2$ values between the GPP observed and GPP MOD17A2H (Table 2) were 0.76 and 0.65 for DC and SC, respectively, which are satisfactory. Between the GPP observed and GPP modeled, the values of $R^2$ were smaller, i.e., 0.28 and 0.12 for DC and SC, respectively. However, the average values were closer to the in-situ measurements. The modeled GPP presented the smallest difference (mean absolute err—MAE) in relation to the ground measurements, showing for the DC, an MAE value of 3.2 g·C·m$^{-2}$·day$^{-1}$, while for the GPP MOD17A2H, the MAE was 4.0 g·C·m$^{-2}$·day$^{-1}$. The modeled GPP presented better results (with smaller absolute errors), since it is calculated by associating in situ measurements with orbital data. However, spatially, the GPP MOD17A2H presented better results, particularly in the differentiation of land cover types between the two study areas.

### 3.6. Spatial Analysis

Remote sensing imagery provides a useful way to capture not only the magnitudes of different biophysical variables but also their distribution throughout a given region. In this study, the spatial distribution of ET and GPP is crucial because they can provide important insights, for example, on how water is being used to produce biomass, or how much carbon is absorbed considering different hydrological conditions in a specific year. This can bring benefits for future studies in the Caatinga region in order to establish priority areas for conservation and also agriculture purposes in the surrounding areas.

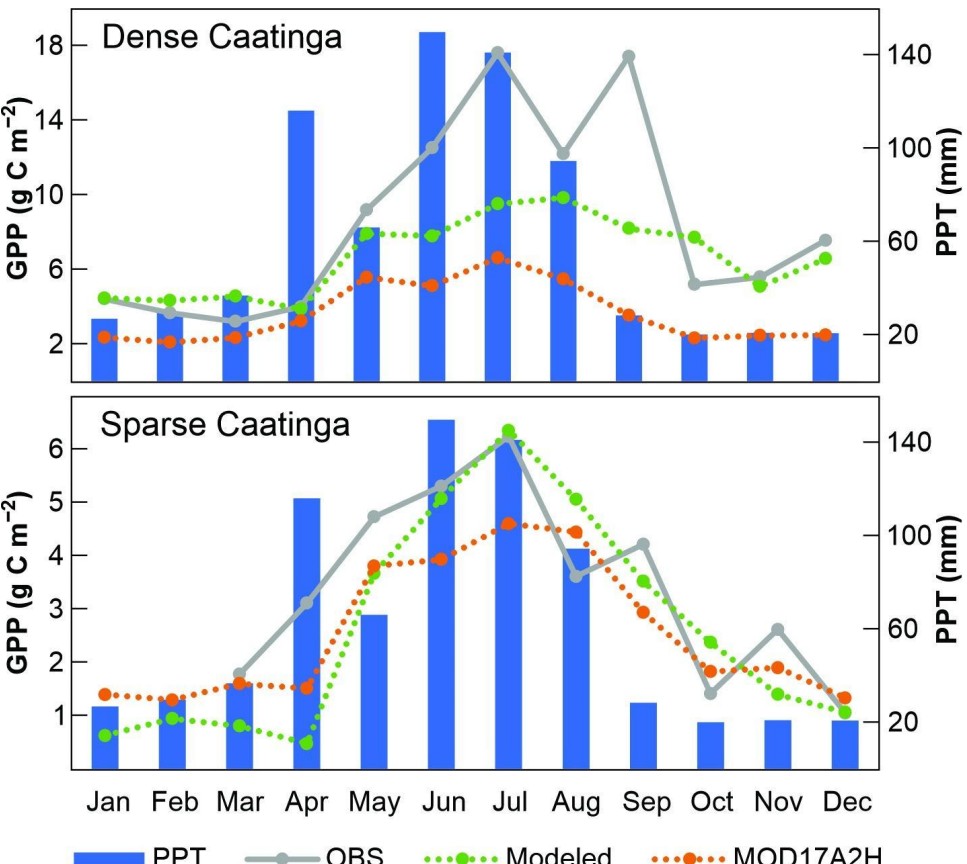

**Figure 3.** Monthly temporal distribution of GPP in the areas of DC and SC in 2013.

The spatial distribution of annual ET from the MOD16A2 and SEBAL products for 2013 in Paraíba is shown in Figure 4. The annual ET ranged from 30 to 1200 mm·year$^{-1}$, with higher values on the coast and western portion of the state. Comparing the two models (MOD16A2 (a) and SEBAL (b)) with the observed data, it is possible to observe an overestimation of the ET derived from SEBAL in the areas that presented values greater than 500 mm·year$^{-1}$. In addition to the annual overestimation, ET SEBAL also presented an overestimation in the dry period, mainly in the DC area (Figure 2). In addition, the authors of [45] showed an overestimation of ET SEBAL (geeSEBAL) in the woodland savanna in the dry period when compared to the observed data. This comparison suggests that the SEBAL model may not accurately represent the actual ET levels in Caatinga regions. The overestimation of ET in SEBAL could be associated with various factors, including limitations in input data quality, incorrect model parameters, and/or limitations in the algorithm used. It is important to continue monitoring and evaluating ET models to ensure they accurately represent the water cycling processes in Caatinga ecosystems and can inform sustainable resource management practices.

The annual cumulative maps of the GPP in 2013 in Paraíba are shown in Figure 5. The values vary between 0 and 3000 g·C·m$^{-2}$·year$^{-1}$. The highest GPP values were observed in the eastern part of the map (Figure 5a), where the Atlantic Forest and the transition to the Caatinga are located. The central part of the map, where the Borborema mesoregion is located, showed low GPP values for the two models, MOD17A2H (a) and Modeled (b). In comparison with the two maps, Figure 5b (modeled) overestimated the GPP values when these values were above 1500 g·C·m$^{-2}$·year$^{-1}$ in the MOD17A2H model (a) and underestimated the values in areas where the GPP was low (<1000 g·C·m$^{-2}$·year$^{-1}$).

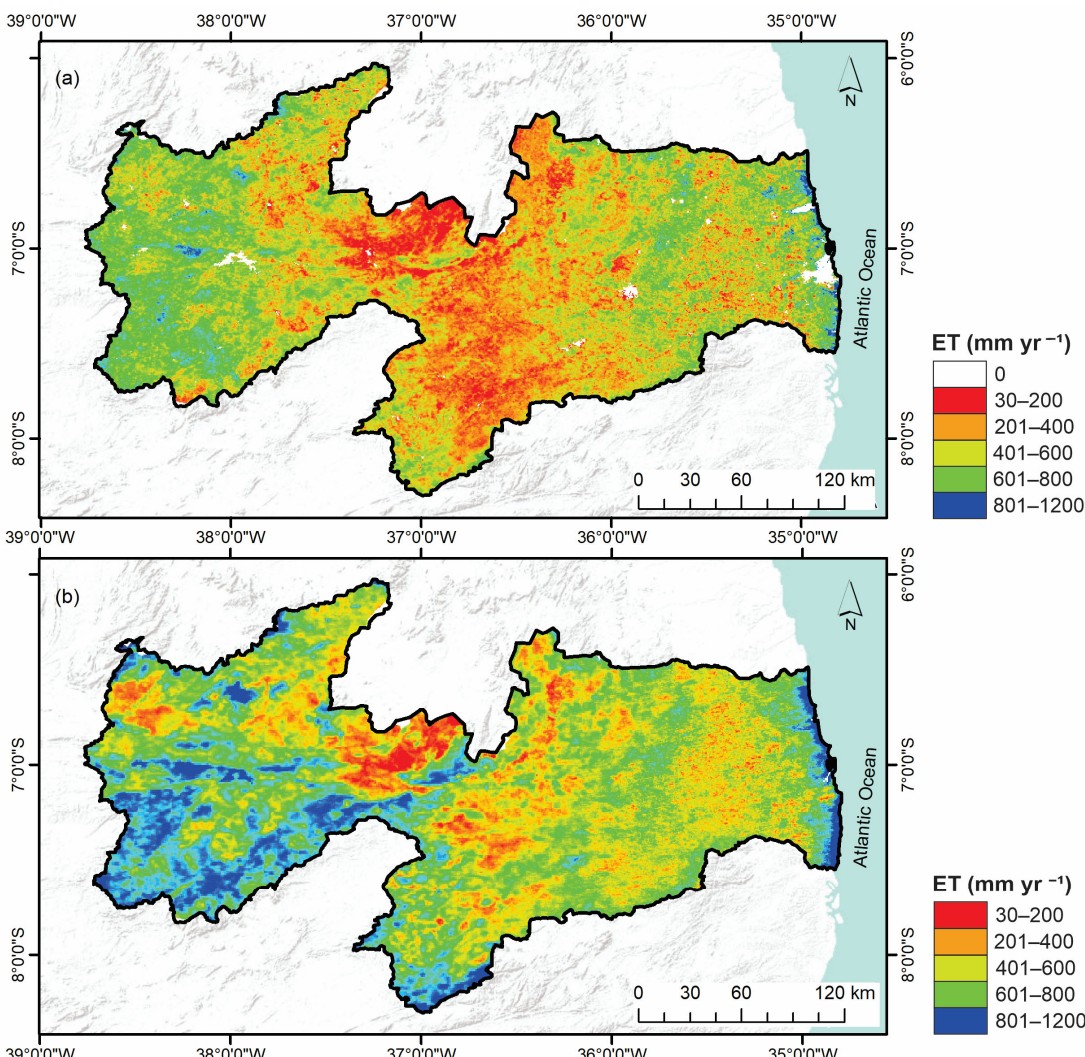

**Figure 4.** Annual ET derived from the products MOD16A2 (**a**) and SEBAL (**b**) for the state of Paraíba for 2013.

There is a greater absorption of $CO_2$ in the eastern portion of the Agreste, with higher GPP values than in the western portion (where the Caatinga vegetation predominates), as was the case for the ET. This can be explained by the fact that the eastern portion is closer to the Zona da Mata, with a more humid climate than the western portion. At Zona da Mata, there is a predominance of Atlantic Forests.

The eastern part of the Agreste showed higher GPP values on all days analyzed, showing the relationship of GPP with soil moisture content. Thus, rainfall within this region seems to be a limiting factor for the absorption of $CO_2$ by the Caatinga, since the dry months have the lowest rates. This factor can be explained by the phenological dynamics of the Caatinga, which quickly responds to greater water availability [13]. According to [10], in dry periods, or even in the rainy season of dry years, there is a decrease in the absorption of $CO_2$ by this biome.

Figure 6 shows the annual spatial statistical analysis (SD, maximum–max, and median) of the ET derived from MOD16A2 for Paraíba in 2013. Figure 6a shows which areas had lower ET rates (Figure 4), which were those that did not suffer major variations throughout the year. Areas such as the Agreste and Sertão of Paraíba showed greater variations in ET throughout the year, with a standard deviation close to 3 mm·day$^{-1}$. These areas that showed greater variations throughout the year were also the areas where the maximum reached close to 10 mm·day$^{-1}$ (Figure 6b), where these values are considered high for the

region. It is possible to observe that the central region of the map reached a maximum of ~1 mm·day$^{-1}$, which makes sense for this region to reach SD close to 0 (Figure 6a), as these are regions where the ET values are very low throughout the whole year. Observing Figure 6c, it is noted that most of the state of Paraíba has a median value of between 0.5- and 1.0-mm·day$^{-1}$, except for the Mata Paraibana mesoregion, where the median values are higher, above 2 mm·day$^{-1}$, reaching values close to 10 mm·day$^{-1}$ in some areas. This happens because it is a region with the presence of the Atlantic Forest biome, where ET rates are higher than in the Caatinga biome as stated in [12].

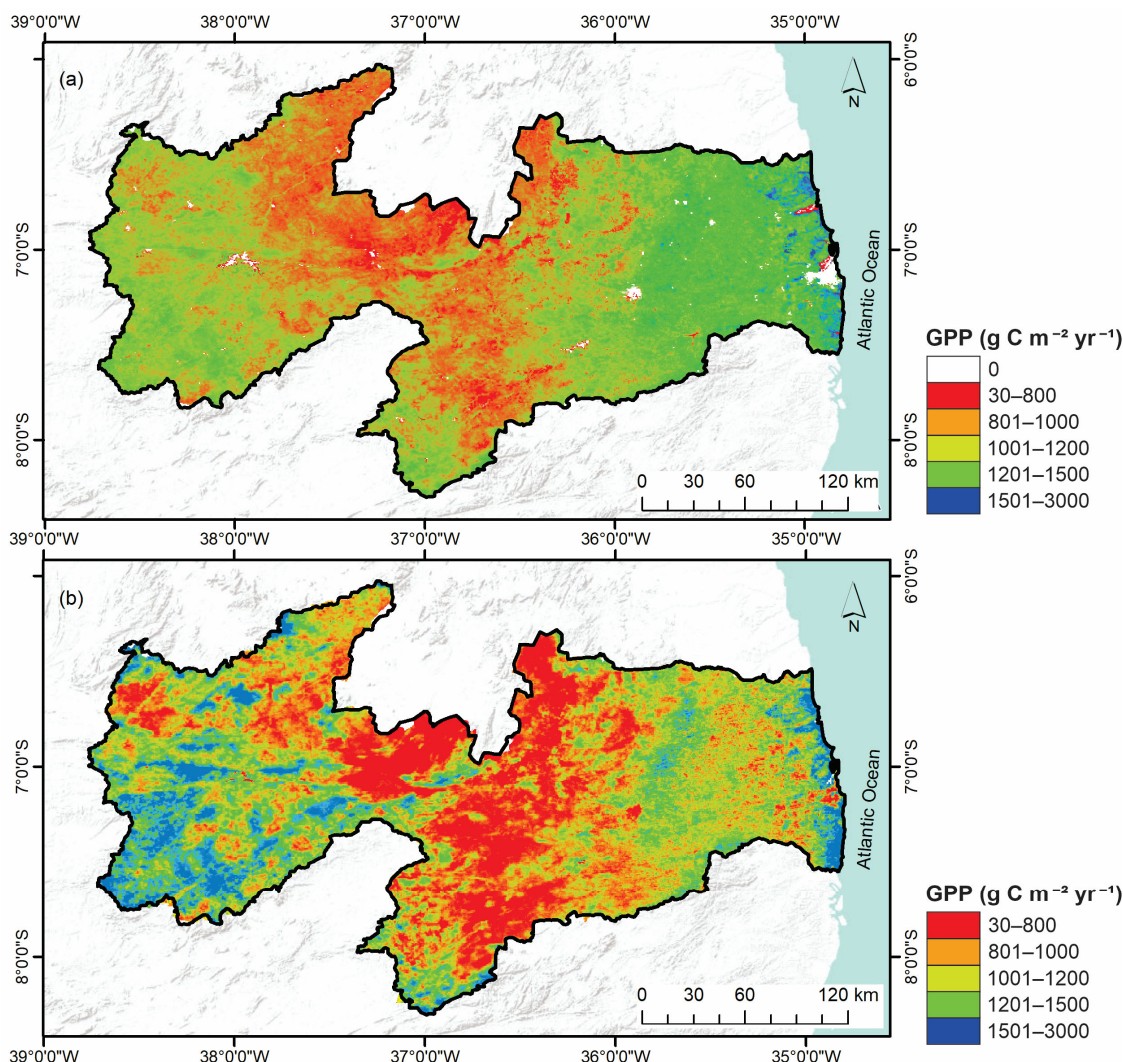

**Figure 5.** Annual GPP derived from the product MOD16A2 (**a**) and Modeled (**b**) for the state of Paraíba for 2013.

Figure 7 shows the annual spatial statistical analysis (SD, maximum–max, and median) of the GPP derived from the MOD17A2H product for Paraíba in 2013. Through the SD map (Figure 7a), it is possible to observe that the central region of Paraíba has a good portion of GPP homogeneity throughout the year, reaching a variation below 1 g·C·m$^2$·day$^{-1}$ throughout the year. The greatest variations occurred in the Agreste and Sertão Paraibano mesoregions, reaching a maximum SD of 4 g·C·m$^2$·day$^{-1}$. These results support the statistical map of the maximums (Figure 7b), which shows that the maximum GPP was low in areas where there were no significant variations in GPP throughout the year and that the maximum was close to 10 mm·day$^{-1}$ in areas where there were significant variations (Agreste and Sertão Paraibano). The map showing the GPP median (Figure 7c) presents an

almost homogeneous map with median values below 2.5 mm·day$^{-1}$, except for the Mata Paraibana mesoregion, where the median was slightly higher, above 5 mm·day$^{-1}$.

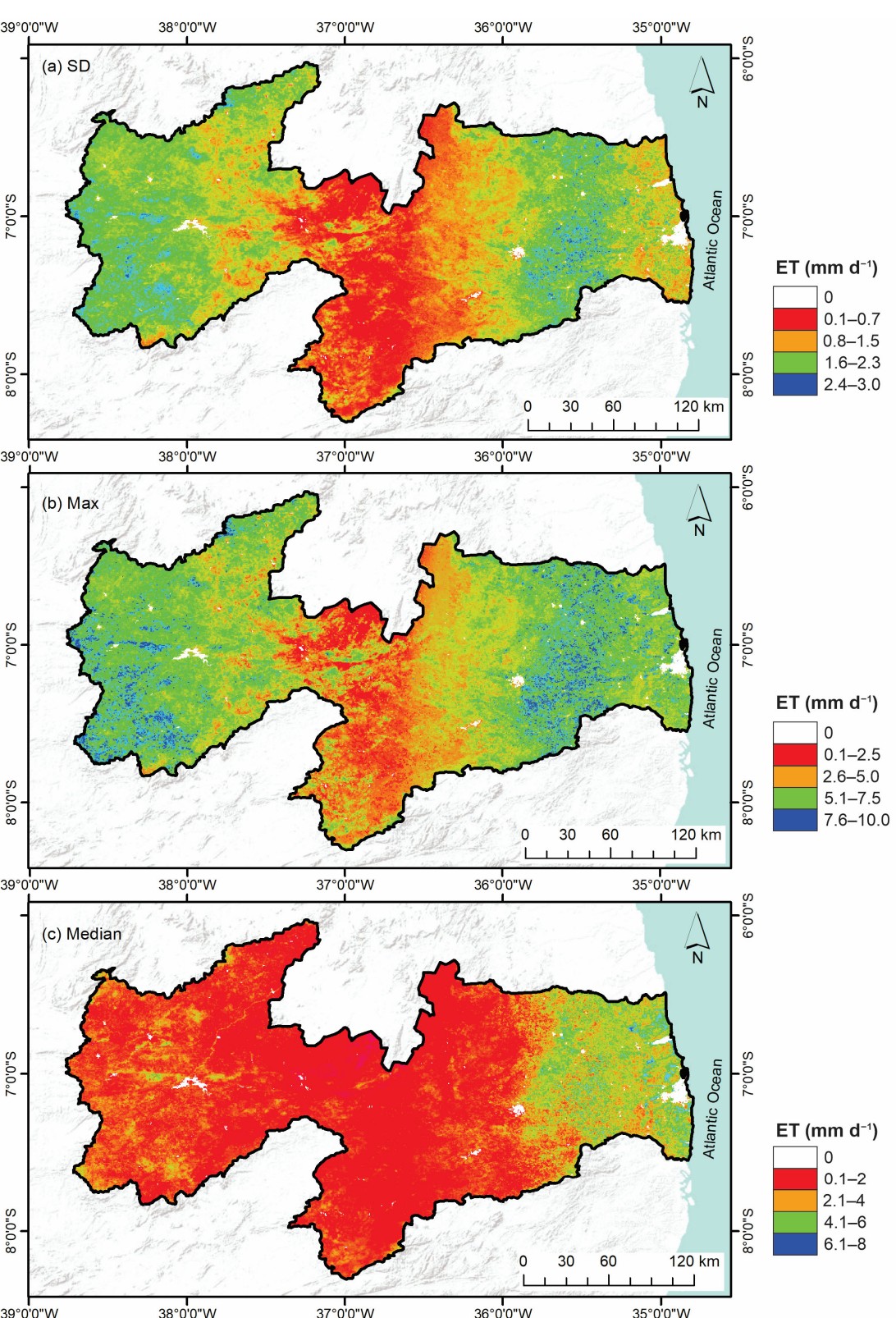

**Figure 6.** Statistical analysis (standard deviation—SD, Maximum—max, and median) of the ET for the state of Paraíba in 2013 derived from the product MOD16A2.

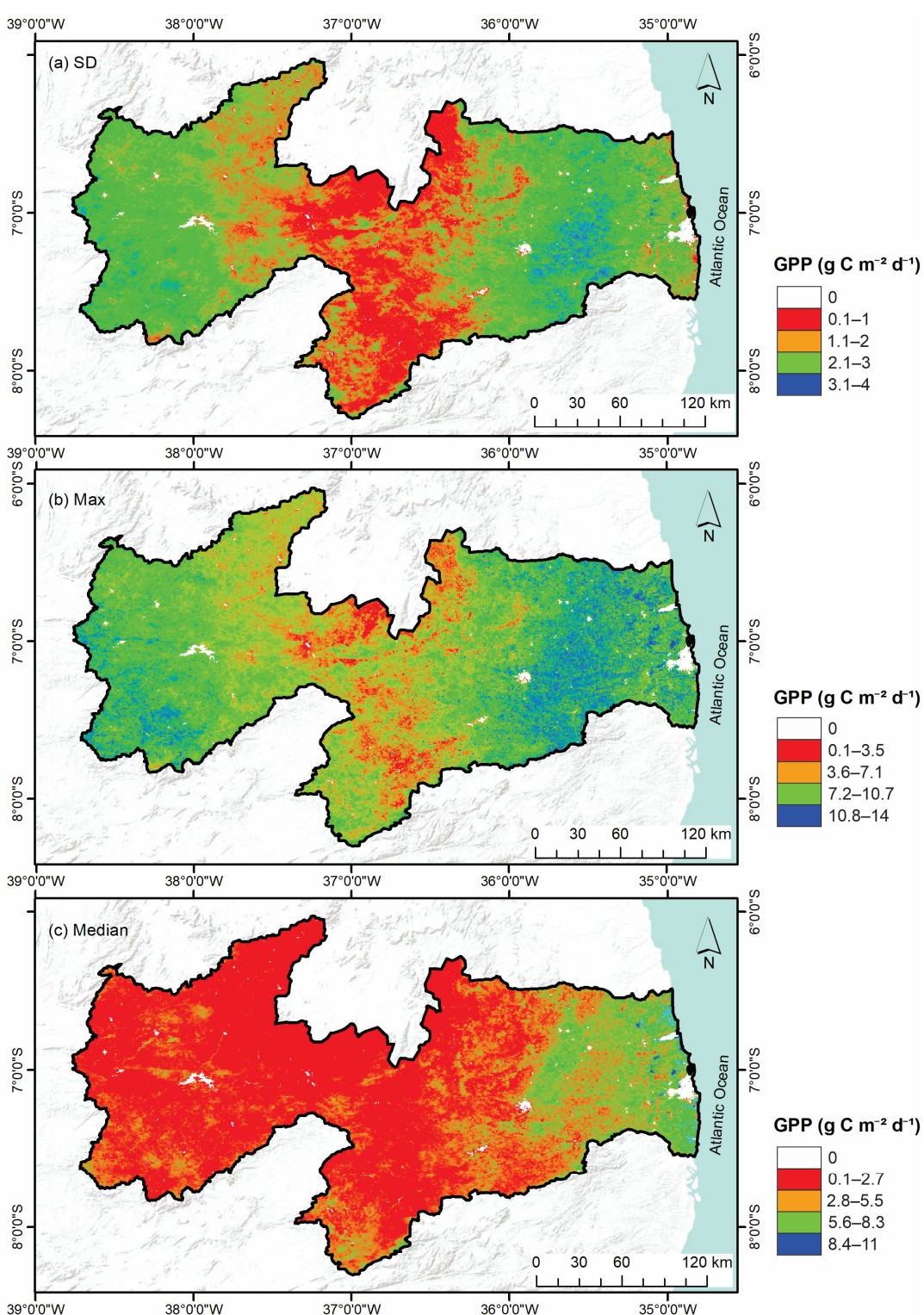

**Figure 7.** Statistical analysis (Standard Deviation–SD, Maximum–max, and median) of GPP for the state of Paraíba in 2013 derived from the product MOD17A2H.

## 4. Conclusions

The energy, water, and carbon fluxes for the Caatinga biome obtained are key to addressing current environmental issues that face this important region. Our results show, among others, the greater availability of energy to carry out physical processes in the area with the greatest amount of vegetation (DC), with higher values for them compared to the

sparse Caatinga (SC). We also found that rainfall is a key factor in the Caatinga's ability to capture carbon and highlight the need for further studies on the effects of climate change on this biome.

The ET also presented higher values when compared to the SC, which indicates that degradation tends to reduce the evapotranspiration process since the removed vegetation will imply less water vapor transferred to the atmosphere.

The influence of water availability on gross primary productivity (GPP) for the regions with preserved Caatinga and regions with degraded Caatinga under study was also found to be evident. These differences were better observed through the estimates via MOD17A2H, which presented a better correlation with the observed data. However, these results were less accurate in comparison to the field measurements than the model proposed in this work (modeled GPP). Rainfall is a key factor in Caatinga's capacity to capture $CO_2$. The greatest differences between the dense and sparse areas occur during higher/lower water availability, depending on the season. We believe that the results found here will help to design new studies focusing on the role of current climate change and its effect on contrasting Caatinga areas, based, for example, on whether they are more or less capable of releasing water and absorbing carbon given their level of degradation. This study can also bring benefits to the Caatinga region by helping to further establish priority areas for conservation and agriculture in the surrounding areas.

Summarizing, this study highlights the importance of preserving the Caatinga biome for its ecological and societal benefits. The results suggest that areas with greater vegetation (DC) have higher availability of energy and water, as well as greater carbon capture capacity. On the other hand, the degradation of the Caatinga biome leads to reduced evapotranspiration and carbon capture. This highlights the need for more sustainable resource management practices. These findings can help prioritize areas for the conservation and sustainable development in the Caatinga region. Moreover, they can serve for the development of priority policies reports, for example, in order to show the involved stakeholders what paths should be considered in order to contribute to the preservation of this important ecosystem.

**Author Contributions:** Conceptualization, M.L.d.O., C.A.C.d.S. and F.A.C.S.; methodology, M.L.d.O., C.A.C.d.S. and F.A.C.S.; software, M.L.d.O. and F.A.C.S.; validation, M.L.d.O. and F.A.C.S.; formal analysis, M.L.d.O. and F.A.C.S.; investigation, M.L.d.O., C.A.C.d.S. and F.A.C.S.; resources, C.A.C.d.S.; data curation, M.L.d.O. and F.A.C.S.; writing—original draft preparation, M.L.d.O., C.A.C.d.S., F.A.C.S., G.d.O., C.A.G.S., U.A.B., J.E.d.B.L.C. and R.M.d.S.; writing—review and editing, M.L.d.O., C.A.C.d.S., F.A.C.S., G.d.O., C.A.G.S., U.A.B., J.E.d.B.L.C. and R.M.d.S.; visualization, M.L.d.O., C.A.C.d.S., F.A.C.S., G.d.O., C.A.G.S., U.A.B., J.E.d.B.L.C. and R.M.d.S.; supervision, C.A.C.d.S.; project administration, C.A.C.d.S.; funding acquisition, C.A.C.d.S. All authors have read and agreed to the published version of the manuscript.

**Funding:** FAPESQ—Fundação de Apoio à Pesquisa do Estado da Paraíba (Edital N° 09/2021 Demanda Universal), Chamada Interna Produtividade em Pesquisa PROPESQ/PRPG/UFPB (Edital N° 04/2021), and the Conselho Nacional de Desenvolvimento Científico e Tecnológico (CNPq) (Process No. 304493/2019-8).

**Acknowledgments:** This study was supported by the FAPESQ—Fundação de Apoio à Pesquisa do Estado da Paraíba (Edital N° 09/2021 Demanda Universal), Chamada Interna Produtividade em Pesquisa PROPESQ/PRPG/UFPB (Edital N° 04/2021), and the Conselho Nacional de Desenvolvimento Científico e Tecnológico (CNPq) funded the second author with a Research Productivity Grant (Process No. 304493/2019-8).

**Conflicts of Interest:** The authors declare no conflict of interest.

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
