# Peer review of "Evaluation of Water and Carbon Estimation Models in the Caatinga Biome Based on Remote Sensing"

_forests, doi:10.3390/f14040828_

Round 1

Reviewer 1 Report (New Reviewer)

Would encourage authors to improve their work by addressing the following suggestions in order for their paper to be considered for publication.

Author Response

Answers to the reviewer's comments

The authors are grateful for the kind words and constructive comments on the article with the new title: “Land degradation and its impacts on energy, water, and carbon fluxes in the Caatinga based on MODIS and ground measurements”, offered by anonymous reviewers. We respond appropriately to all comments, as you can find below. In the revised manuscript, all changes are highlighted in red. We express profound gratitude for your review, which has significantly improved the quality of the manuscript.

Reviewer #1

I reviewed the article and must say that even some findings are promising; the article lacks novelty and some key representation of the results.

Thank you very much for the positive feedback. We agree with the reviewer that the paper needs to improve the clarity of the text to highlight its scientific contributions.

The introduction is skimpily written and omits a thorough, current literature assessment. Additionally, it doesn't specify where gaps between the present study and earlier studies are filled.

Even though the authors employed the MODIS product to assess the ET, there are a number of studies that have identified biases that the authors have overlooked, therefore it would be wonderful to add those studies. Additionally, give a thorough justification of how the bias in the MOD16A2 was fixed. I advise including these more references because they are extremely pertinent and will give their introduction more scientific credibility. (https://doi.org/10.1061/(ASCE)IR.1943-4774.0001199; https://doi.org/10.1016/j.agwat.2016.01.006).

Response: We have taken measures to improve the introduction of our paper by providing a comprehensive literature review and defining the gaps that our study aims to fill. We have also incorporated additional references to provide greater scientific credibility to our work. (Lines 95-113)

The authors list a number of research that make use of MODIS to measure evapotranspiration. However, the results or recommendations in terms of their modelling techniques are not given.

Elaborate more on how this paper differs or affirms the findings and conclusion of those studies. These points need to be clearly addressed in the introduction section.

Methodology is simplistic and not well described. For instance, while ET estimate using MODIS is good, there are a number of flaws that need to be fixed or standardised, and information about those flaws is not provided in paper.

Response: With regards to the methodology section, we have included more details on the differences and similarities of our study with previous studies that have used MODIS to measure evapotranspiration. We have also clarified the limitations and standardizations of the MODIS data in our estimation of ET.

So authors have used GPP but nowhere they have show the temporal pattern of GPP alongwith EVI and ET, to know they behave to seasonality. Seasonal variation in terms of wet period and dry period should be presented in the study. Perhaps more importantly authors should also show how if there is any hysteresis pattern seen or not.

Response: Regarding the representation of results, the seasonality of the GPP is shown in Figure 3, where the GPP is compared with the PPT. For the analysis of the ET seasonality, we compared the water loss with the vegetation index (Figure 2).

How author have MODIS product with observed data, the foot print should be shown in the methodology.

Response: We add a studied footprint for MODIS products. (176-182)

Why authors rely on EVI provide justification?

Response: Studies indicate that the EVI is more efficient in identifying the shrubby Caatinga, taking into account variations in seasonality, exposure, and terrain inclination, as this index presents the soil adjustment factor for this type of biome.

Thank you once again for your feedback. We have made the necessary revisions to enhance the quality and clarity of our paper.

Reviewer 2 Report (New Reviewer)

The study tries to provide the land degradation and its impacts on energy, water, and carbon fluxes based on MODIS and ground measurements. The following is the specific suggestions:

1. The article should offer line numbers for clear review.

2. The contents of article is not matching with the title. The title is about land degradation and its impacts on energy, water, and carbon fluxes. However, the majority of content is about the comparison of different calculation methods and the accuracy of results. The validation of results is too much, and the discussion about land degradation and its impacts should be added to the article.

3. ET may be the keyword for this article.

4. Formula 9 about NDVI is not correct.

5. If the measurements performed in SC must have had some interference that caused an increase in the observed Rn, the observed Rn24 may be not accurate.

Author Response

Answers to the reviewer's comments

The authors are grateful for the kind words and constructive comments on the article with the new title: “Land degradation and its impacts on energy, water, and carbon fluxes in the Caatinga based on MODIS and ground measurements”, offered by anonymous reviewers. We respond appropriately to all comments, as you can find below. In the revised manuscript, all changes are highlighted in red. We express profound gratitude for your review, which has significantly improved the quality of the manuscript.

Reviewer #2

The study tries to provide the land degradation and its impacts on energy, water, and carbon fluxes based on MODIS and ground measurements. The following is the specific suggestions:

  1. The article should offer line numbers for clear review.

Line numbers have been inserted in the article.

  1. The contents of article is not matching with the title. The title is about land degradation and its impacts on energy, water, and carbon fluxes. However, the majority of content is about the comparison of different calculation methods and the accuracy of results. The validation of results is too much, and the discussion about land degradation and its impacts should be added to the article.

We agree that the article works more on evaluating models in the studied area, so we decided to change the title of the article to: "Evaluation of water and carbon estimation models in the Caatinga biome based on remote sensing". In addition, we add a more in-depth discussion of soil degradation and its impacts.

  1. ET may be the keyword for this article.

The word evapotranspiration was added to the keywords.

  1. Formula 9 about NDVI is not correct.

Formula 9 has been fixed.

  1. If the measurements performed in SC must have had some interference that caused an increase in the observed Rn, the observed Rn24 may be not accurate.

Borges et al. (2020) showed that, in the SC area, the incidence of solar radiation was higher. These results indicate that the high solar radiation in this site may be a crucial contributing factor to the higher values of Rn.

Round 2

Reviewer 1 Report (New Reviewer)

 Accept in present form

Author Response

Thanks for your valuable comments.

Reviewer 2 Report (New Reviewer)

The title is not changed in the article.

Author Response

Done in the new version. Thanks for showing us the mistake.

This manuscript is a resubmission of an earlier submission. The following is a list of the peer review reports and author responses from that submission.

Round 1

Reviewer 1 Report

Authors present in their manuscript a study that investigated the differences in the dynamics of critical environmental variables such as net radiation (Rn), evapotranspiration (ET), and carbon fluxes (GPP) in contrasting recovered Caatinga (dense Caatinga) and degraded Caatinga (sparse Caatinga) in the state of Paraíba, northeastern Brazil.

I think the paper is well written and it is neatly exposed. The literature cited is adequate and so are the graphics. Therefore, I consider this paper suitable for publication on Forest. However, reading the paper I have had the feeling that the description, in particular for the discussion of the results, is overly prolific. Probably it need to be summarised in a better and clear way. The abstract also should be reduced.

I suggest a revision of the manuscript according to few minor comments (in the file attached) before to consider it for publication on the Journal.

Author Response

All corrections are described in the attached manuscript. 

Reviewer 2 Report

This manuscript compares energy (net radiation), water (ET) and carbon (GPP) measured at Eddy covariance towers, estimated from MODIS images, and modelled (only for ET and GPP) for two contrasted Caatinga sites in Brazil, one site feature by dense woody vegetation and the other with more sparse vegetation. While the manuscript provides useful information, unfortunately the description of the system and two study areas is rather poor:

·        What the size of the footprint for the EC covariance stations?

·        What is the vegetation cover within each footprint? Type, cover, LAI …?

·        Same information should be provide for the MODIS pixel, to assess to what extent  EC data represent the vegetation of the MODIS pixels.

While reading the section 2.4 on data processing, it was not fully evident when authors referred to MODIS data and when to modelled data, later presented in Table 2 and Figures 2 and 3. I would suggest more clarity on this to get better less skilled readers.

Section 3.6, on results of the spatial analysis, remains to basic, descriptive, based on the visual comparison of maps. I think this deserve a more statistical comparison.

Same comment applies for the discussion of the results. Results are poorly discussed from a scientific point of view. Authors just used some references to confirm that they results are within the range of data previously published, but barely explain the reason of differences among Dense and Sparse sites, and neither why MODIS and model perform reasonable well in some site/season but not in the others. As both MODIS and models show important deviations respect to the measures data, authors should also discuss some ideas to advance, to improve the reliability of MODIS and models for the study systems.

There are some general conclusions that are quite questionable or need more nuances. For instance, in lines 5-7 of page 13, it says “This evidences the fact that our methodology is a reliable alternative and simple to perform for estimating ET in the Caatinga biome, both for the dry and rainy seasons”. But according to figure 2 this is valid for DS site but not at all for the SD site. And in this regard, I do not understand quite well why in Table 2 estimated values are higher than observed values and then Figure 2 shows the contrary.

Also in page 13, authors state that “ET SEBAL showed this (ET) information more accurately”, but SEBAL is overestimating a lot ET, and barely follows the seasonal variability observed.

The use of literature is also poor and rather outdated. For instance, to emphasize the lack of scientific knowledge for the Caatinga ecosystem, authors refer to 2002 reference (20 years ago!!!). Moreover, there are lot of studies performed in other savannoids and/or semiarid ecosystems across the world that could results of high interest for the discussion.

Please, check text referring to under or overestimation. Sometimes seems changed (e.g. L1-3 in page 11: As a result, the ET estimated according to Equation 12, here called ET SEBAL, and ET according to the MOD16 product, were underestimated in relation to the measured data (estimated by the EC technique). According to Table 2, this is right only for ET MOD16A2 at SC, but the contrary for the other three combinations.

Other details

Please, use WP instead W in eq. 3 for precipitable water (WP is used later on).

Please, check eq. 7. LE term is lacking. Please, check also eq. 22

Simplify the title of section 2.4.3.

Author Response

All corrections are in the attached file.

Round 2

Reviewer 2 Report

I appreciate the authors' efforts to provide a little more information, but I am afraid it is still insufficient. The main shortcomings were not addressed or resolved.

Looking at Figure 1, the 1 km2 area that surrounds the DC EC is occupied by both Savanna and Caatinga; for SC the situation is even more complex, with urbanised areas within the 1 km2 footprint area. If the vegetation homogeneity of the 90-95% footprint cannot be confirmed, it would be important to filter out the data coming from the specific areas from the total data (for each time, the more contributing area can be estimated). Otherwise you are introducing heterogeneity in the data that may not correspond to what the chosen MODIS pixels provide. I still think it is very important to provide a detailed map of the vegetation present in the 1 km2 footprint of each EC tower and of the MODIS pixels used to compare with the EC results.

Since the area of greatest contribution to the EC station might vary over the calendar, the MODIS pixels used might also have to be different, so that the vegetation represented by both sources (EC and MODIS) would be more similar.

In any case, according to table 2, the measured GPP values correlate very poorly with those modelled and in absolute terms differ greatly from those provided by MODIS. As shown in figure 3 for ET, SEBAL performed very badly in both sites, and MOD16A2 also performed very badly in SC. Also, according to figure 3, modelled and MODIS-based GPP values badly followed the observed seasonally. The major shortcomings of the MODIS products and modelled results are discussed, but not adequately explained, and no possible solutions are provided to move forward. Thus, the mere detection of deficiencies (and some acceptable correlations) is of little use and of little scientific value.

As I discussed in the previous review, Figures 4 and 5 show huge differences in absolute terms for MODIS-measured and modelled ET and GPP values. These differences are commented visually, but not analysed (e.g., under which land uses the results deviate the most). Thus, the discussion of the results is of little value.

Consequently, I cannot recommend the publication of this work.

Author Response

Answer to Reviewer 2

Dear Reviewer,

We would like to thank you for reading, reviewing, and evaluating this work, which will undoubtedly contribute to improvements in the quality of our text. Thus, we present below the answers to the questions raised and some suggestions.

Comments and Suggestions for Authors

I appreciate the authors' efforts to provide a little more information, but I am afraid it is still insufficient. The main shortcomings were not addressed or resolved.

Looking at Figure 1, the 1 km2 area that surrounds the DC EC is occupied by both Savanna and Caatinga; for SC the situation is even more complex, with urbanised areas within the 1 km2 footprint area. If the vegetation homogeneity of the 90-95% footprint cannot be confirmed, it would be important to filter out the data coming from the specific areas from the total data (for each time, the more contributing area can be estimated). Otherwise you are introducing heterogeneity in the data that may not correspond to what the chosen MODIS pixels provide. I still think it is very important to provide a detailed map of the vegetation present in the 1 km2 footprint of each EC tower and of the MODIS pixels used to compare with the EC results.

Answer: Figure 1 has been changed (Page 4). Following Borges et al. (2020) (Pages 4–5), the high-frequency sensors (10 Hz) that compose the EC system include a three-dimensional sonic anemometer (CSAT3A, Campbell Scientific, Inc., Logan, UT, USA) to measure the three wind speed components (u, v, w), and a gas analyzer (EC150, Campbell Scientific, Inc., Logan, UT, USA) to measure the water vapor (H2O) and carbon dioxide (CO2) concentrations. The instruments were installed following the user guide described by Campbell Scientific Inc. (2020), pointing to the predominant wind direction (east) to minimize flux distortions by the analyzer supporting arms or other components (Campos et al., 2019; Mendes et al., 2020). A series of corrections were carried out in the collected signals for air temperature fluctuations and air density (WPL) (Webb et al., 1980; Foken, 2008; Aubinet et al., 2012), and the coordinate rotation (2D rotation) was applied directly to the CRBasic language program of the CR3000 data loggers. In addition to the periodic and careful maintenance of instruments, data were submitted for a rigorous post-processing method. For the 30-min averages of variables such as wind speed, turbulent fluxes, and temperatures, among others, a filter was applied to remove them. 

Since the area of greatest contribution to the EC station might vary over the calendar, the MODIS pixels used might also have to be different, so that the vegetation represented by both sources (EC and MODIS) would be more similar.

Answer: Done (Page 4)

In any case, according to table 2, the measured GPP values correlate very poorly with those modelled and in absolute terms differ greatly from those provided by MODIS. As shown in figure 3 for ET, SEBAL performed very badly in both sites, and MOD16A2 also performed very badly in SC. Also, according to figure 3, modelled and MODIS-based GPP values badly followed the observed seasonally. The major shortcomings of the MODIS products and modelled results are discussed, but not adequately explained, and no possible solutions are provided to move forward. Thus, the mere detection of deficiencies (and some acceptable correlations) is of little use and of little scientific value.

Answer: Done. All text is in red in the results section.

As I discussed in the previous review, Figures 4 and 5 show huge differences in absolute terms for MODIS-measured and modelled ET and GPP values. These differences are commented visually, but not analysed (e.g., under which land uses the results deviate the most). Thus, the discussion of the results is of little value.

Answer: Done in Section 3.6.
